# A Combination of In Silico ADMET Prediction, In Vivo Toxicity Evaluation, and Potential Mechanism Exploration of Brucine and Brucine N-oxide—A Comparative Study

**DOI:** 10.3390/molecules28031341

**Published:** 2023-01-31

**Authors:** Yan Gao, Lin Guo, Ying Han, Jingpu Zhang, Zhong Dai, Shuangcheng Ma

**Affiliations:** 1National Institutes for Food and Drug Control, Beijing 100050, China; 2Department of Pharmacology, Institute of Medicinal Biotechnology, Chinese Academy of Medical Sciences and Peking Union Medical College, Beijing 100050, China

**Keywords:** brucine, brucine N-oxide, ADMET, zebrafish, network pharmacology

## Abstract

Brucine (BRU) and brucine N-oxide (BNO) are prominent, bioactive, and toxic alkaloids in crude and processed Semen Strychni. Studies have demonstrated that BRU and BNO possess comprehensive pharmacological activities, such as anti-inflammatory and analgesic. In this context, a comparative study of BRU and BNO was performed by combination analysis of in silico ADMET prediction, in vivo toxicity evaluation, and potential action mechanism exploration. ADMET prediction showed that BRU and BNO might induce liver injury, and BRU may have a stronger hepatoxic effect. The prediction was experimentally verified using the zebrafish model. The BRU-induced hepatotoxicity of zebrafish larvae had a dose-response relationship. The mechanism of BRU-induced hepatotoxicity might relate to phosphorylation, kinase activity, and signal transduction. By comparison, signal transduction and gap junctions might involve BNO-induced hepatotoxicity. Our results provided a better understanding of BRU- and BNO-induced hepatotoxicity. We also built a foundation to elucidate the material base of the hepatotoxicity of traditional Chinese medicine Semen Strychni.

## 1. Introduction

Brucine (BRU) and brucine N-oxide (BNO) are bioactive and toxic constituents isolated from Semen Strychni. It is traditional Chinese medicine and has been mainly used to treat several diseases such as rheumatoid arthritis, cancer pain, and myasthenia due to its analgesic and anti-inflammatory activities [1,2,3,4,5,6]. BRU, the second most abundant alkaloid in Semen Strychni, is white crystalline powder soluble in menthol. The study demonstrated that BRU was highly distributed in blood-supply tissues after intravenous administration of crude alkaloid fractions of possessed Semen Strychni to rats. The highest content of BRU was found in kidney, followed by small intestine, spleen, liver, lung, stomach, heart, and plasma [7]. Besides anti-inflammatory and analgesic activities, BRU was reported to inhibit various tumor cells, including liver, skin, mammary, lung, gastric, and colorectal cancer lines, based on diverse physiological mechanisms [8,9,10,11,12,13,14,15]. New drug delivery systems of BRU have been a hot area of research in recent years. The release rate and time of BRU in vitro are controlled to play a long-term and effective medicinal effect and have an excellent clinical application prospect [16]. However, high toxicity has limited its clinical application. BRU is concerned with neurotoxicity, hepatotoxicity, nephrotoxicity, cytotoxicity, and cardiac toxicity [17,18,19,20]. A recent study showed that brucine hepatotoxicity was dosing time-dependent in mice caused by circadian Cyp3a11 metabolism [17]. Researchers are committed to reducing its toxicity and improving its therapeutic index. Therefore, further investigating the toxic effect and mechanism of brucine is significant.

BNO can be partly transformed from brucine during the processing of Semen strychni, which is considered a potential mechanism for processing detoxification [21]. The transformation also happens in vivo and in vitro [7,21,22,23,24]. A study on the contrast found that BNO can convert to BRU in vitro [25]. BNO, one of the major alkaloids in processed Semen Strychni, was reported to have equally potent pharmacological and less toxic side effects than BRU [26]. Several studies [1,26,27] showed BNO possessed anti-inflammatory, analgesic, and antiplatelet aggregation activities. In addition, it could help to control excessive alcohol behavior. BNO increased the safety margin, as LD50 of BNO is nearly 1/15 of BRU by intraperitoneal administration in rats [28]. Moreover, BNO exhibited no activity on hERG channels, while BRU inhibited it and induced cardiac toxicity [17]. However, there are few reports on the toxicity of BNO.

Drug toxicity research is essential to ensure medication safety. As for the main ingredients of natural medicines, a toxicity study provides evidence to elucidate the mechanism of toxicity and discover potentially unknown adverse reactions. Up-to-date data show that hepatotoxicity is one of the major causes for failure in drug development, drug withdrawal, and limiting drug clinical application [29,30,31]. Over the past few years, the zebrafish model has been increasingly employed in pre-clinical hepatotoxicity research in vitro. Numerous studies [32,33] have been published on the changes in liver size or liver degeneration in treated zebrafish larvae and this suggests that the drug is hepatotoxic. Compared to wild-type zebrafish, a transgenic line of zebrafish that expressed enhanced red fluorescent protein (dsRed) provided more precise and intuitive results because the liver was fluorescent. With the development of computer technology and chemoinformatics, various cost-effective and time-saving computational tools are applicated to predict ADMET (absorption, distribution, metabolism, elimination, and toxicity) properties in a high-throughput manner, based on the structure-activity/structure-property relationship [34,35,36].

The present study aimed to test the hypothesis of the hepatoxicity of BNO as compared to its parent drug BRU. For this purpose, ADMET predictor in silico was used to obtain unreported druggability information, complement the reported experimental results, and predict their potential toxicity. A transgenic zebrafish line with liver-specific dsRed expression in the liver was used to validate the predictions and evaluate BRU-and BNO-induced liver injury. Furthermore, our study preliminarily explored the mechanism of BRU and BNO using network pharmacology.

## 2. Results and Discussions

### 2.1. Calculation of ADMET-Related Properties

With the web server ADMETlab, physicochemical properties, medicinal chemistry, and ADMET properties of BRU and BNO were calculated (Table 1). Comments on the value of each indicator were shown in the Appendix A. There is a correlation between the first two programs that several physicochemical properties are critical components of medicinal chemistry. TPSA is a valuable parameter for predicting drug mobility, which is significantly related to the small intestinal absorption of the drug in the human body, the permeability of Caco-2 monolayer, and the penetration of the blood-brain barrier. The TPSA value of BNO is slightly higher than that of BRU, as both are optimal. LogS value represents the water solubility index of small molecule compounds, while logP value reflects the distribution of substances in the two phases of oil and water. LogD (PH 7.4) value is usually used to express the apparent partition coefficient of a drug in the intestinal environment. All of the logS, logP, and logD values of BNO and BRU are in the optimal range. Based on the specific values, it can be inferred that the water solubility of BNO is stronger than BRU’s. QED value is used to assess drug-likeness by comprehensively analyzing critical physical and chemical properties compared with approved drugs, which is a crucial consideration when selecting compounds early in drug discovery. The SAscore value represents the ease of synthesis of compounds. Both the QED and SAscore values of BRU are ideal. In contrast, BNO is not. 

As for predicted results of ADMET properties, most notably, BRU- and BNO-induced human hepatotoxicity (H-HT) may exist, as category 1 represents H-HT positive(+) and category 0 represents H-HT negative(-). As BRU is predicted to be a substrate of P450 CYP1A2, CYP3A4, CYP2C19, and CYP2D6, it may have stronger hepatotoxicity than BNO. A previous study [19] has shown that mouse Cyp3a11, an orthologous gene of human CYP3A4, which was believed to perform similar functions as CYP3A4, was a significant contributor to brucine metabolism. Based on the results of absorption, BRU and BNO are probably easily absorbed orally. BRU and BNO may be P-glycoprotein substrates and can be actively effluxed via p-glycoprotein. According to another study, it was compatible with the result that a highly effective P-gp modulator candidate should possess at least one tertiary basic nitrogen atom [37]. BRU may have good Caco-2 Permeability. BRU was predicted to be a P-gp inhibitor, widely considered a powerful and effective method to reverse multidrug resistance (MDR) in cancer. BRU was probably easier to cross the blood–brain barrier.

This study used computer simulation for the ADMET prediction of BRU and BNO, which was short time-consuming, feasible, effective, and practical. However, the prediction value could only be analyzed by comparing with the reference range, which required more validation studies. Further experimental research should be carried out at the cellular and animal level.

### 2.2. In Vitro Embryotoxicity Test for the Prediction of Acute Zebrafish Toxicity to BRU and BNO

Wild-type zebrafish embryos were exposed to different concentrations of BRU and BNO, and embryonic death was observed under the microscope. The results were shown in Figure 1 that the LC50 of BRU is 0.25 mM, and the LC50 of BNO is 6.33 mM.

### 2.3. BRU- and BNO-Induced Hepatoxicity in Zebrafish

#### 2.3.1. Effects of BRU and BNO on the Liver Phenotype of Wild-Type Zebrafish

As shown in Figure 2, BNO had no significant toxic effects at low concentrations (0.01 mM) compared to the control group and exhibited concentration-dependent liver blackening and slightly increased liver area at 0.05 mM–0.1 mM. However, the difference was not significant. The hepatotoxicity of brucine was observed in the zebrafish model in this study, which is consistent with the previous research findings [19] that brucine has hepatotoxicity in the mouse model. The BRU-induced hepatotoxicity of zebrafish larvae had a dose-response relationship. That is, with the increase of the administration concentration, the liver darkened gradually and the liver area gradually increased. The positive control acetaminophen did not affect the size of the zebrafish larval liver within the administration range. However, the liver also gradually turned black with the increase of the administration concentration, indicating liver toxicity.

#### 2.3.2. Effects of BRU and BNO on Liver Development in Transgenic Zebrafish

Compared to the non-administered control group (Figure 3), the high concentration of BRU and BNO in the administration group (0.1 mM) could induce significant changes in liver morphological changes in transgenic zebrafish larvae. The enlarged fluorescence area and/or the enhanced fluorescence intensity of the liver indicated that the expression of the liver-type fatty acid-binding protein-related gene fabp10a was enhanced, and the liver area was enlarged. However, the positive control drug acetaminophen did not significantly affect the liver of zebrafish larvae within the dose range.

### 2.4. Potential Action Mechanism Exploration of BRU and BNO

The functional and signaling pathway enrichment analysis of BRU- and BNO-related gene clusters were performed by GO and KEGG, respectively.

#### 2.4.1. Functional Enrichment Analysis

In order to clarify the biological actions, a functional enrichment analysis was performed, and the enrichment results were displayed in order of significance, as shown in Figure 4 and Figure 5. The abscissa in the figure represents the number of genes enriched in the entry, and the ordinate represents the entry name. The redder color indicates greater significant biological action. It reflects that one compound may regulate multiple complex biological processes to exert multiple pharmacological effects.

As for BRU, the top 10 remarkable significant biological processes were chosen according to the corrected FDR from small to large, involving protein phosphorylation, phosphorylation, signal transduction, peptidyl-serine phosphorylation, protein autophosphorylation, peptidyl-tyrosine phosphorylation, intracellular signal transduction, peptidyl-threonine phosphorylation, and other processes.

Protein phosphorylation is a fundamental process of cellular biological effects, and many diseases are characterized by an imbalance in the activities of both protein kinases and phosphatases. Protein phosphorylation also plays a vital role in the regulation of cell proliferation. Kinases are also the primary regulators of cancer. Members of the receptor tyrosine kinase family are involved in the occurrence and regulation of many cancers. Studies showed that BRU might exert an antitumor effect via several mechanisms.

As for BNO, the top 10 biological processes were signal transduction, response to xenobiotic stimulus, phospholipase C-activating G-protein coupled receptor signaling pathway, sensory perception of pain, regulation of synaptic vesicle exocytosis, Positive regulation of ERK1 and ERK2 cascade ERK1, G-protein coupled receptor signaling pathway, and others.

G protein-coupled receptors (GPCRs) are the largest class of membrane proteins in the human genome, which are the targets for most drugs [38]. Both opioid receptors and 5-HT (except for 5-HT3) receptors are GPCRs. Serotonin (5-HT) is a monoamine and widely distributes in the nervous system as a neurotransmitter. 5-HT receptors’ agonists and antagonists have been reported to have therapeutic action in neuropathic pain [39]. Besides being closely related to therapeutic pain relief, opioid receptors are involved in various pleiotropic functions, including but not limited to cell proliferation, epileptic seizures, and immune function [40]. Functions related to GPCRs, especially opioid receptors and 5-HT receptors, may be the mechanism of the analgesic effect of BNO.

#### 2.4.2. Pathway Enrichment Analysis

The enrichment results are displayed in order of significance, as shown in Figure 6 and Figure 7. The abscissa in the figure represents the number of genes enriched in the pathway, and the ordinate represents the pathway name. The redder color indicates greater significant pathways.

A total of 135 pathways (*p* < 0.05) were obtained for BRU, mainly involving PD-L1 expression and PD-1 checkpoint pathway in cancer, prostate cancer, neuroactive ligand-receptor interaction, progesterone-mediated oocyte maturation, T cell receptor signaling pathway, Yersinia infection, Fc epsilon RI signaling pathway, calcium signaling pathway, endocrine resistance, phospholipase D signaling pathway, and other biological processes.

A total of 22 pathways (*p* < 0.05) were obtained from BNO, mainly involving neuroactive ligand-receptor interaction, calcium signaling pathway, endocrine resistance, estrogen signaling pathway, bladder cancer, pathways in cancer, relaxin signaling pathway, gap junction, proteoglycans in cancer, cocaine addiction, and other biological processes.

#### 2.4.3. “Component-Target-Pathway” Network of BRU and BNO

The “component-target-pathway” network of BRU is composed of 72 nodes (61 compound target nodes, 10 pathway nodes and 1 compound nodes) and 155 edges, as shown in Figure 8. The network of BNO contains 40 nodes (29 compound target nodes, 10 pathway nodes and 1 compound nodes) and 78 edges, as shown in Figure 9. The core target genes of BRU mainly included MAPK1, AKT1, EGFR, RAF1, MAPK3, etc. The core target genes of BNO mainly included MMP9, EGFR, ESR1, MDM2, MMP2, etc.

According to the physiological processes that have been reported to be associated with drug-induced liver injury (DILI), further analysis of functional and pathway enrichment results associated with BRU- and BNO-induced hepatotoxicity was performed.

Protein kinases, which catalyze the phosphorylation of proteins, contribute to antagonizing or promoting liver injury [41]. Researchers have found that APAP increased the expression of the phosphorylation of NFκB p65 (Ser536) and JNK (Thr183/Tyr185). At the same time, endoplasmic reticulum stress-induced upregulation of STARD1 could promote APAP-induced acute liver failure via SH3BP5 and phosphorylation of JNK1 and JNK2 [42,43]. Troglitazone-induced hepatotoxicity is related to the activation of JAK/STAT pathway [44]. As overwhelming oxidative stress plays a critical role in DILI, signal transduction pathways activated/inhibited during oxidative stress proved to be one of the critical factors in DILI [45,46]. Many signal transduction pathways have been found involved in developing APAP hepatotoxicity, such as JNK, Mtor, and NFκB [47]. Among the top 10 functions and pathways related to BRU, biological actions involved in BRU-induced hepatotoxicity may include phosphorylation, kinase activity, and signal transduction.

Same as BRU, signal transduction may be involved in BNO-induced hepatotoxicity. In terms of the pathway, gap junctions should be concerned. Recent studies found that dysfunction of gap junctions is also involved in various liver pathological processes, such as acute liver injury [48,49].

## 3. Materials and Methods

### 3.1. Prediction of ADMET In Silico

A freely accessible web-based ADMET predictor platform ADMETlab 2.0 (https://admetmesh.scbdd.com/, accessed on 12 May 2022) was applicated on the ADMET prediction of BRU and BNO. It is one of the most comprehensive platforms. It provides a computational model supporting the calculation of 88 ADMET-related endpoints, including 17 physicochemical properties, 13 medicinal chemical properties, 23 ADME properties, 27 toxicity endpoints, and 8 toxic groups [50].

### 3.2. Toxicity Evaluation on Zebrafish

#### 3.2.1. Test Compounds and Solution Preparation

Acetaminophen (APAP) was used as positive control chemical. Brucine (110706–201306, 91.7%) and acetaminophen (100018–202111, 99.9%) were national reference substances obtained from the National Institutes for Food and Drug Control (Beijing, China). BNO was prepared in our lab. The structure of BNO was confirmed by NMR and MS. The purity of BNO (>95%) was normalized by HPLC. The structures of brucine and brucine N-oxide were shown in the Appendix A. Stock solutions and dilutions of all compounds were prepared before use in freshly prepared zebrafish feeding solution and stored at −20 °C. Dosing solutions of BRU and BNO yielded concentrations of 0.05, 0.1, 0.2, 0.5, 1, 4, and 8 mM in the zebrafish embryotoxicity test. As for the zebrafish hepatotoxicity test, the dosing solutions of BRU and BNO were 0.01, 0.025, 0.05, and 0.1 mM, while APAP dosing concentrations were 1, 8, and 16 mM, respectively.

#### 3.2.2. Zebrafish Embryo Toxicity Test (ZET)

Wild-type zebrafish (Danio rerio, TU strain) were used to perform a zebrafish embryo toxicity test. Zebrafish maintenance and embryo collection for the test according to the guideline of OECD was described in the Appendix A. Embryos (20 embryos/dish) at 6 h post-fertilization (6 hpf) stage were immersed in 2 mL of each dosing solution in a 20 mm dish and allowed to develop under standard conditions until 96 hpf. The control groups were treated with embryo water. During this period, the embryonic state was observed by an Olympus SZX16 stereomicroscope (Tokyo, Japan) and recorded every day. A series of concentrations (0.05, 0.1, 0.2, 0.5, 1, 4, 8 mM) were applied to find a narrower concentration range. The BRU-induced mortality was between 0% and 100%, as the BNO was 0%–80%, considering it is hard-obtained. The experiment was repeated at least three times. Mortality is expressed as mean and standard deviation (Mean ± SD).

#### 3.2.3. Zebrafish Hepatotoxicity Test (ZHT)

ZHT was initiated at 72 hpf and terminated at 120 hpf. A transgenic zebrafish line with liver-specific dsRed expression was used to assess the potential hepatotoxicity of the two chemicals. Wild-type and transgenic line of zebrafish larvae at 72 hpf (20 larvae in each group) were exposed to BRU and BNO at 0.01, 0.025, 0.05, and 0.1 mM for a 2-day treatment period, respectively. Transgenic zebrafish larvae (72 hpf) in the positive control group of APAP were dosed at 1, 8, and 16 mM for 2 days. Larvae at 120 hpf (at least 10 larvae in each group) were taken to record the liver morphology under a fluorescence microscope. The liver size of zebrafish larva was quantified using Image J software. The experiment was repeated three times. 

The liver toxicity of BRU and BNO was determined by measuring the liver size changes, which was calculated as follows: Liver size (%) = liver area (drug)/liver area (control) × 100%.

### 3.3. Network Pharmacology-Based Exploration of Potential Action Mechanism 

#### 3.3.1. Identification of Putative Targets of BRU and BNO

100 potential targets associated with BRU and 40 potential targets related to BNO were screened from the SwissTargetPrediction database (http://www.swisstargetprediction.ch/, accessed on 7 July 2022), with the species set as “mouse”. The Gene Symbols were retrieved from the UniProt database (https://www.uniprot.org/, accessed on 8 July 2022) corresponding to the target name.

#### 3.3.2. GO and KEGG Pathway Enrichment Analysis

GO (gene ontology) is a comprehensive database describing gene function for systematically analyzing gene function at the molecular and cellular levels. It can be divided into three parts: Molecular function (MF), biological process (BP), and cellular component (CC). KEGG (kyoto encyclopedia of genes and genomes) is a public database resource for high-level functions and biological systems (such as cells, organisms and ecosystems), integrating genomic, chemical and system functional information. The categories of KEGG are represented by signaling pathways. GO enrichment analysis and KEGG pathway analysis were performed using Database Visualization and Integrated Discovery system (DAVID) (https://david.ncifcrf.gov/home.jsp, accessed on 15 July 2022).

#### 3.3.3. Construction of “Component-Target-Pathway” Network

Cytoscape 3.7.2 software was used to construct a “component-target-pathway” network to obtain the relevant pathways and targets of BRU and BNO, respectively.

### 3.4. Statistical Analysis

The experimental results were described as mean ± SD. The concentration-effect curves of compounds were drawn using GraphPad Prism 8.0. SPSS software was used for statistical data and statistical significance was determined based on one-way ANOVA analysis. The LSD method was used to compare the two groups.

## 4. Conclusions

The results provided by the ADMET predictor indicated that both BRU and BNO might induce liver injury. Moreover, the probability of inducing hepatotoxicity of BRU is greater than that of BNO. The hepatotoxicity of BRU and BNO in wild-type and transgenic zebrafish showed a dose-dependent relationship. The BRU-induced hepatotoxicity in zebrafish is greater than that of BNO. Through network pharmacology analysis, it was found that potential mechanisms of BRU-induced hepatotoxicity might relate to phosphorylation, kinase activity, and signal transduction. While signal transduction and gap junctions might be involved in BNO-induced liver injury.

## Figures and Tables

**Figure 1 molecules-28-01341-f001:**
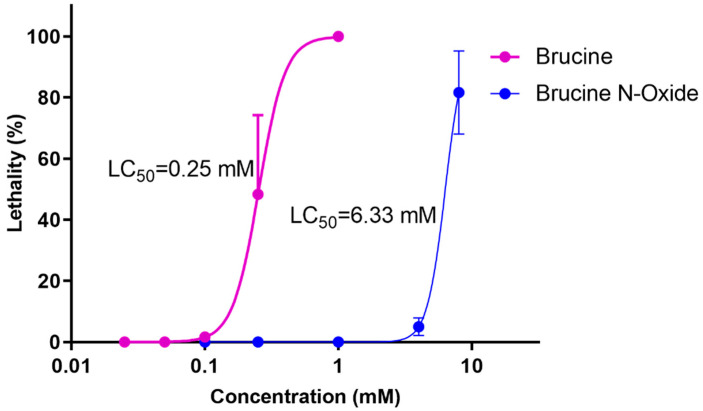
Dose-response curves of BRU- and BNO-induced acute toxicity in zebrafish.

**Figure 2 molecules-28-01341-f002:**
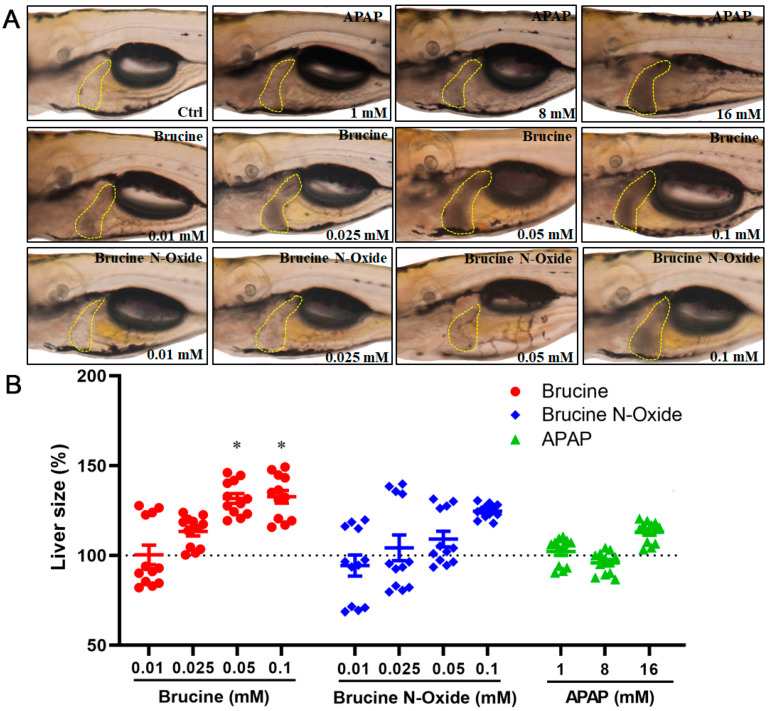
Effects of BRU and BNO on the liver phenotype of wild-type zebrafish. (**A**) Abnormal phenotypes caused by BRU, BNO and APAP on wild-type zebrafish, (**B**) Comparison of toxicity of BRU, BNO and APAP in wild-type zebrafish. Values represent the mean ± SEM (n = 20), * *p* < 0.05.

**Figure 3 molecules-28-01341-f003:**
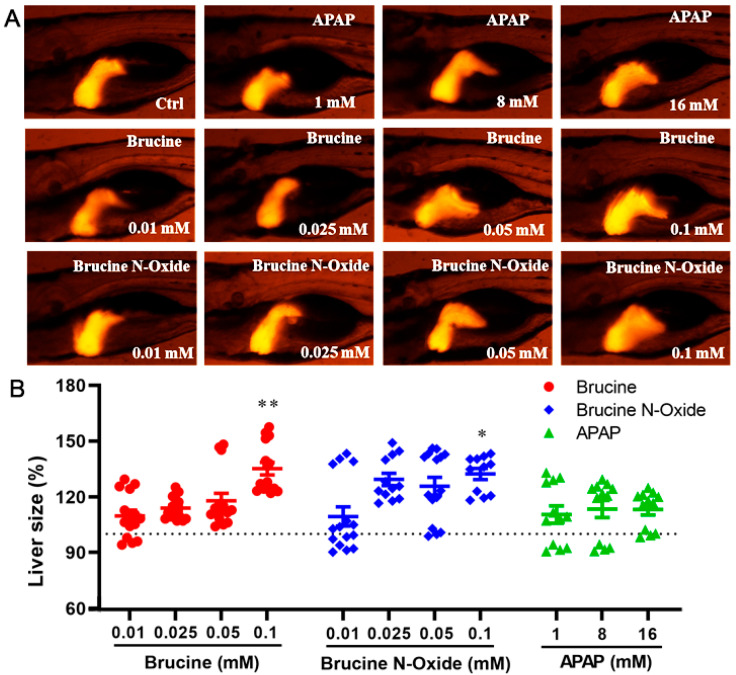
Effects of BRU and BNO on liver development in transgenic zebrafish. (**A**) Abnormal phenotypes caused by BRU, BNO and APAP on transgenic zebrafish, (**B**) Comparison of toxicity of BRU, BNO and APAP in transgenic zebrafish. Values represent the mean ± SEM (n = 20), * *p* < 0.05, ** *p* < 0.01.

**Figure 4 molecules-28-01341-f004:**
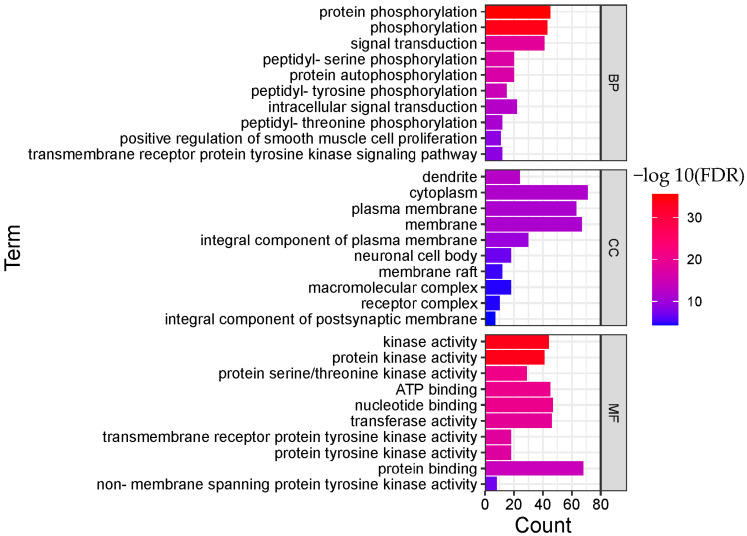
Functional enrichment analysis result of BRU (TOP 10; BP, biological process; CC, cellular component; MF, molecular function).

**Figure 5 molecules-28-01341-f005:**
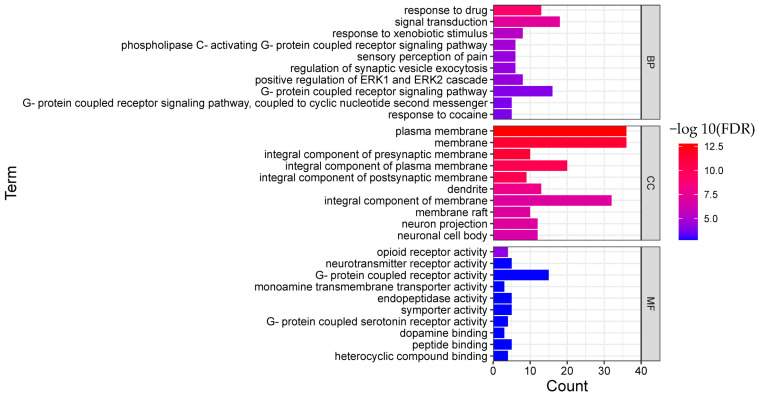
Functional enrichment analysis result of BNO (TOP 10; BP, biological process; CC, cellular component; MF, molecular function).

**Figure 6 molecules-28-01341-f006:**
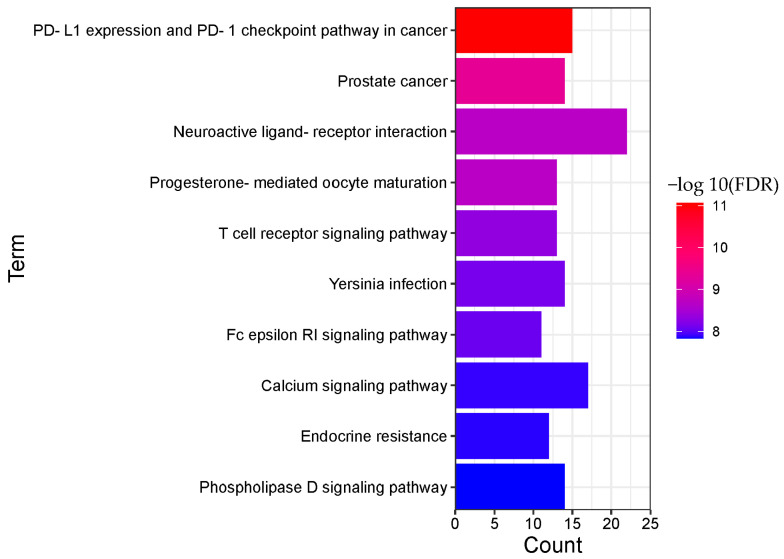
Pathway enrichment analysis result of BRU (TOP 10).

**Figure 7 molecules-28-01341-f007:**
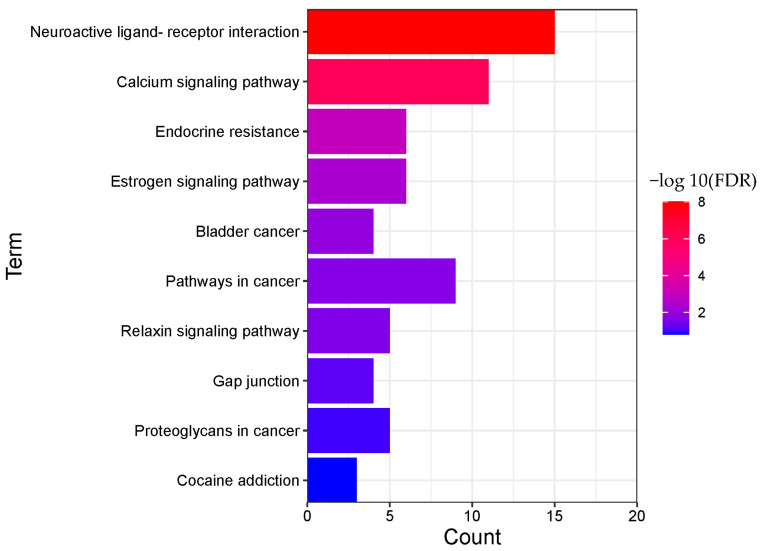
Pathway enrichment analysis result of BNO (TOP 10).

**Figure 8 molecules-28-01341-f008:**
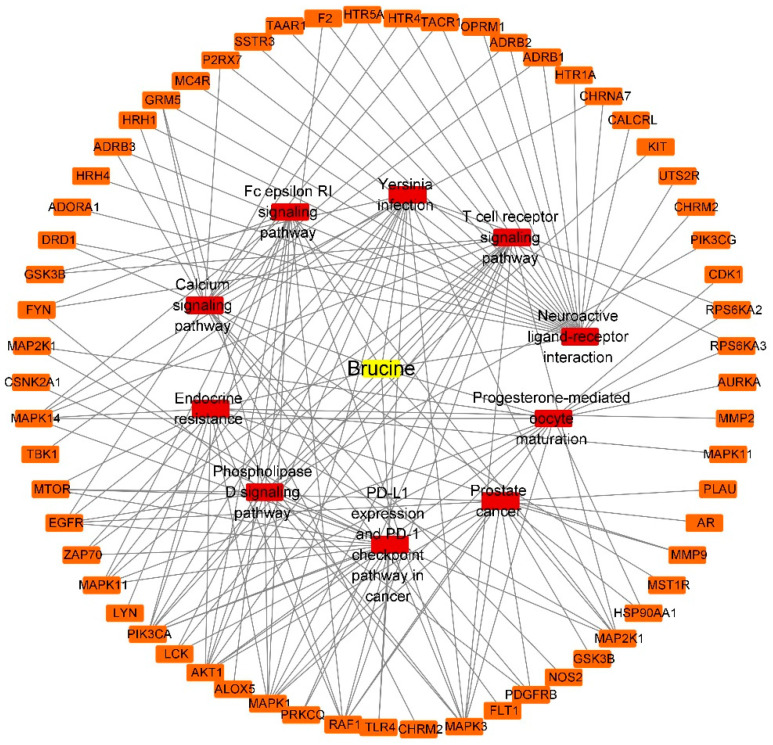
“Component-target-pathway” network of BRU (Yellow: compounds; red: pathways; orange: genes).

**Figure 9 molecules-28-01341-f009:**
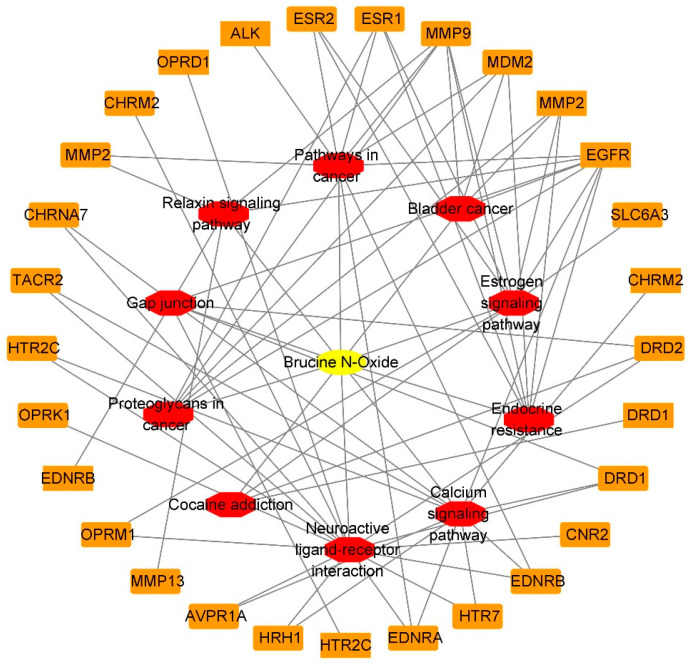
“Component-target-pathway” network of BNO (Yellow: compounds; red: pathways; orange: genes).

**Table 1 molecules-28-01341-t001:** Predicted values of physicochemical, medicinal chemical, and ADMET properties of BRU and BNO.

Property	Predicted Values	
Physicochemical Property	Brucine	Brucine N-Oxide
TPSA	51.24	71.06
LogS (Solubility)	−3.126 log mol/L	−1.702 log mol/L
LogD7.4 (Distribution Coefficient D)	2.083	0.392
LogP (Distribution Coefficient P)	1.585	1.443
Medicinal Chemistry		
QED	0.719	0.424
SAscore	5.739	6.162
Absorption		
Papp (Caco-2 Permeability)	−4.609	−5.312
Pgp-inhibitor	0.96	0.001
Pgp-substrate	0.611	0.998
HIA (Human Intestinal Absorption)	0.011	0.908
Distribution		
PPB (Plasma Protein Binding)	48.53%	22.48%
VD (Volume Distribution)	1.483 L/kg	1.518 L/kg
BBB (Blood–Brain Barrier)	0.952	0.314
Metabolism		
P450 CYP1A2 inhibitor	0.024	0.011
P450 CYP1A2 Substrate	0.764	0.683
P450 CYP3A4 inhibitor	0.221	0.029
P450 CYP3A4 substrate	0.887	0.920
P450 CYP2C9 inhibitor	0.026	0.008
P450 CYP2C9 substrate	0.191	0.147
P450 CYP2C19 inhibitor	0.026	0.018
P450 CYP2C19 substrate	0.914	0.827
P450 CYP2D6 inhibitor	0.023	0.002
P450 CYP2D6 substrate	0.532	0.274
Elimination		
T 1/2 (Half Life Time)	0.699 h	0.928 h
CL (Clearance Rate)	10.81 mL/min/kg	11.487 mL/min/kg
Toxicity		
hERG (hERG Blockers)	0.034	0.022
H-HT (Human Hepatotoxicity)	0.334	0.368
AMES (Ames Mutagenicity)	0.013	0.009
SkinSen (Skin sensitization)	0.157	0.217
DILI (Drug Induced Liver Injury)	0.187	0.075

## Data Availability

Not applicable.

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
