# Peer review of "A Combination of In Silico ADMET Prediction, In Vivo Toxicity Evaluation, and Potential Mechanism Exploration of Brucine and Brucine N-oxide—A Comparative Study"

_molecules, 2023, doi:10.3390/molecules28031341_

Round 1
Reviewer 1 Report
The manuscript title "A combination of in silico ADMET prediction, in vivo toxicity 2 evaluation, and potential action mechanism exploration of 3 brucine and brucine N-oxide–a comparative study" speaks to the content, however, I don't know if the journal has a word limit for the manuscript. I suggest that this should be looked at to make the title more concise.
The abstract is okay, it has the essential components of an abstract.
The introduction is detailed with relevant references.
The methodology is adequately described with the use of up-to-date techniques.
The results were well synthesized with appropriate tables and figures.
The discussion should be improved to include previous studies and establish a relationship between the study and previous studies.
The conclusion is adequate.
Author Response
Thanks to your valuable advice, the title of the paper has been streamlined, and the discussion section has been improved to increase the relationship between this study and previous research. Thank you again for your guidance and help.
Reviewer 2 Report
In table1 page 4 starting with row 183 there is no comments on the colored values , red are not good , green are good values and yellow are??? And the values with now colors are ???? Are they easy to interpret and understand? No!!! These datas are not interpreted appropriately and consistently throughout the manuscript. Please include details regarding this . It was presented the specific database that was used but this one is not working for me. May be in this moment there is a problem
The interpretation of the results must be improved. They are significant. A lot of results but not so much interpretations , Why not?
Please make much discussion for the results obtauned

Author Response
Thank you for your valuable advice. Your understanding is correct that red is a bad value, green is a good value, and yellow is a value with a bad tendency. The ADMETlab prediction platform gives a good or bad reference range for each indicator, and the color is correlated with the result value. Due to space constraints, relevant details of important parameters have been added in the paper according to your suggestions. Details of all indicators will be presented in the supplementary materials.
Reviewer 3 Report
The presented manuscript investigated the hepatotoxicity profile of the Chinese medicine Semen Strychni-bioactive molecules, brucine and brucine N-oxide, through comparative computational pharmacokinetic analysis and in-vivo zebrafish model. Molecular aspects for the compound’s liver injury was further explored through network pharmacology approach. Both gene-function enrichment analysis and constructed component-target-pathway have managed to partially explain the superiority of the brucine-induced hepatotoxicity over its potential metabolite, brucine N-oxide.
This work can be considered as valuable addition to the field of validating the adverse effects of Chinese medicine-isolated metabolites. Publication is recommended following minor suggestions as they are listed below:
1. Chemical structures of brucine and brucine N-oxide should be presented allowing the reader to track the comparative findings, particularly through ADME analysis.
2. In Section 2.2.1, authors are advised to explain the rational for using the adopted concentrations of brucine and positive control acetaminophen. Reference should be adopted.
3. In Figures 2 and 3, brief explanations within the footnote should be presented annotating the main findings, sample sizes, and adopted significance levels.
4. Table 1, accepted/recommended ranges or values for the investigated ADME/Tox parameter should be annotated at the table footnote. Additionally, the presented color codes for the obtained values should be explained.
5. Figure 1. Brucine-N-oxide curve did not reach its own plateau, the thing that would question the accurate estimation of its LC50 value.
6. No post hoc analysis was applied within different analysis. Multiple comparisons test (Tukey or Bonferroni test, among others) should be applied among investigated groups to deeply analysis the pre-clinical data.
7. Steps for chemical synthesis of brucine-N-oxide should be mentioned and structural elucidation data should be placed within the supplementary materials.
8. Authors cited their supplementary materials, yet I could not track the link.
9. In Section 2.3.1, could the authors explain the rational for setting species, particularly as mouse?
10. Authors are advised to elaborate more on the findings obtained from the component-target pathway analysis, where they could not just correlate results with hepatoxicity, yet further correlate findings with reported pharmacological activities of brucine and close-related compound. This could provide extra validity for the obtained network pharmacology analysis.
11. Typos should be revised like; line 196 “mM” instead of “Mm”, and “Semen Strychni” should be italic.
Author Response
Thank you for your valuable advice. According to your suggestions, the article has been modified. The reply is as follows:
- The structure of brucine and brucine N-oxides has been added in the supplementary materials.
- In Section 2.2.1, the concentrations of brucine and positive control acetaminophen in the hepatotoxicity study were determined based on the concentration range when hepatotoxicity was observed in zebrafish in the preliminary experiment.
- The footnotes of Figure 2 and Figure 3 have been supplemented with sample size and adopted significance level. Due to space constraints, the main findings are described in the text.
- The acceptable/recommended ranges or values for the ADME/Tox parameters investigated in Table 1 are supplemented in the supplementary material. The ADMETlab prediction platform gives a good or bad reference range for each indicator, and the color is correlated with the result value. Details of all indicators were presented in the supplementary materials.
- Thank you for your suggestion. Due to the limited quantity of brucine nitrogen oxide samples, it will take at least 6 months to obtain sufficient samples again, so the curve in Figure 1 does not reach the plateau, which is a pity. Fortunately, it does not affect the results of this study on the liver toxicity of brucine N-oxides.
- This is a good proposal, and we will carry out relevant work in other follow-up studies.
- The chemical synthesis steps were carried out with reference to the literature(Cai, B. C., Wu, H., Wang, T. S., Miyashiro, H., & Hattori, M. (1998). Semisynthesis of strychnine N-oxide and brucine N-oxide. JOURNAL OF CHINESE PHARMACEUTICAL SCIENCES, 7, 169-170.).
The analytical data of the structure were as follows: MS m/z: 411.19146 [M+H]+. 13C NMR ppm:25.1(C-15),30.3(C-14),38.9 (C-17),42.2(C-11),47.8 (C-13),53.4(C-7),56.3(C-MeO),56.5(C-MeO),58.7(C-8),64.2(C-23),67.6(C-18),70.7(C-20),75.6(C-12),82.1(C-16),100.7(C-4),105.4(C-1),ll9.9(C-6),134.2(C-21),l35.3(C-22),l35.4(C-5),146.7(C-2),149.9(C-3),168.5(C-lO).
- There may be some problems in the uploading process of supplementary materials. The modified version will be uploaded again this time.
- In Section 2.3.1, the species of mouse is selected because the species of the prediction website can only be selected from Homo sapiens, Mus musculus, Rattus norvegicus. In order to obtain more search results, the species of mouse is selected in this study.
- Thank you for your noble comments, have added the relevant content in the text.
- Thank you for your valuable comments and the corresponding content has been revised.